# Recognition of *Conus* species using a combined approach of supervised learning and deep learning-based feature extraction

**Noshaba Qasmi, Rimsha Bibi, Sajid Rashid** *

National Center for Bioinformatics, Quaid-i-Azam University, Islamabad, Pakistan

* sajid@qau.edu.pk

**Data Availability Statement:** All relevant data are within the paper and its Supporting Information files.

## Abstract

Cone snails are venomous marine gastropods comprising more than 950 species widely distributed across different habitats. Their conical shells are remarkably similar to those of other invertebrates in terms of color, pattern, and size. For these reasons, assigning taxonomic signatures to cone snail shells is a challenging task. In this report, we propose an ensemble learning strategy based on the combination of Random Forest (RF) and XGBoost (XGB) methods. We used 47,600 cone shell images of uniform size (224 x 224 pixels), which were split into an 80:20 train-test ratio. Prior to performing subsequent operations, these images were subjected to pre-processing and transformation. After applying a deep learning approach (Visual Geometry Group with a 16-layer deep model architecture) for feature extraction, model specificity was further assessed by including multiple related and unrelated seashell images. Both classifiers demonstrated comparable recognition ability on random test samples. The evaluation results suggested that RF outperformed XGB due to its high accuracy in recognizing *Conus* species, with an average precision of 95.78%. The area under the receiver operating characteristic curve was 0.99, indicating the model's optimal performance. The learning and validation curves also demonstrated a robust fit, with the training score reaching 1 and the validation score gradually increasing to 95 as more data was provided. These values indicate a well-trained model that generalizes effectively to validation data without significant overfitting. The gradual improvement in the validation score curve is crucial for ensuring model reliability and minimizing the risk of overfitting. Our findings revealed an interactive visualization. The performance of our proposed model suggests its potential for use with datasets of other mollusks, and optimal results may be achieved for their categorization and taxonomical characterization.

## Introduction

Conus Linnaeus is a large genus of gastropods that has been well-preserved in fossil records since its first appearance about 55 million years ago in the Lower Eocene. Cone snails are major predators in tropical reef communities [1, 2]. Their venom contains a diverse array of small peptides (conotoxins) that target neuromuscular receptors and are extensively utilized in

**Funding:** This work has been supported by Higher Education Commission, Pakistan via grant No. 20-15051/NRPU/R&D/HEC/2021. The funders had no role in study design, data collection and analysis, decision to publish, or preparation of the manuscript.

**Competing interests:** The authors have declared that no competing interests exist.

drug development [3–5]. Taxonomic classification of the highly similar cone shell patterns is challenging due to variations in size, color, and geographical distribution. In particular, some Conus species exhibit nearly identical morphological characteristics, making identification difficult and requiring researchers to spend more time on differential analysis. To address these challenges, there is a pressing need to develop more sophisticated computational algorithms or models to automate Conus species recognition and streamline taxonomic classification.

In recent years, due to technological advancements, artificial intelligence (AI) and machine learning (ML) models have emerged as ideal solutions for image recognition [6]. ML algorithms are routinely used to perform various tasks, including pulmonary embolism segmentation via computed tomographic (CT) angiography [7], polyp detection through virtual colonoscopy or CT during colon cancer diagnosis [8], breast cancer detection through mammography [9], brain tumor segmentation using magnetic resonance (MR) imaging [10], and the detection of brain cognitive states through functional MR imaging for diagnosing neurological disorders [11, 12]. ML techniques, such as feature selection and classification, have become crucial for the accurate and automatic diagnosis and prognosis of various brain diseases [13, 14]. For instance, Ronneberger et al. utilized a Convolutional Neural Network (CNN) and data augmentation techniques, achieving promising results by training on an image dataset [15]. Ke et al. proposed a method to enhance the spatial distribution of hue, saturation, and brightness in X-ray images (as image descriptors) to identify unhealthy lung tissues using Artificial Neural Network-based heuristic algorithms [16]. Jaiswal et al. employed Mask-Region-based CNN, a deep neural network approach, which utilizes both global and local features for pulmonary image segmentation, combined with image augmentation, dropout, and L2 regularization for pneumonia identification [17]. Wozniak and Połap simulated the X-ray image inspection process to identify infected tissue locations [18].

Hu et al. used gene eigenvalues and MRI imaging, together with a genetic-weighted random forest (RF) model, to identify key genetic and imaging biomarkers for diagnosis and personalized treatment [19]. Jing et al. applied RF to optical sensors for foreign object debris detection, crucial for aerospace safety [20]. Chen et al. optimized chemical exchange saturation transfer MRI by analyzing frequency contributions using a permuted RF model [21]. Wang and Zhou improved soil organic matter estimates by combining multitemporal Sentinel-2A imaging with RF to benefit agricultural practices [22]. Matese et al. highlighted the role of unmanned aerial vehicle-based hyperspectral imaging in advancing crop health monitoring and management [23]. Barrett et al. emphasized the importance of predictive models in early Huntington's disease intervention [24]. Waldo-Benitez et al. demonstrated ML's impact on enhancing glioblastoma diagnosis and treatment planning through MRI analysis [25]. Huang et al. showed how stacked models improve wheat quality control using hyperspectral imaging [26]. Feng et al. emphasized the need for accurate plume injection height measurements to improve smoke exposure estimates during Australian wildfires [27]. Grandremy et al. provided insights into zooplankton monitoring through advanced imaging in a 16-year Bay of Biscay study [28]. Nobrega et al. applied deep transfer learning to classify lung nodule malignancy [29]. Philips and Abdulla proposed a method for detecting honey adulteration using hyperspectral imaging and ML, enhancing classification models with a feature-smoothing technique [30]. Tao et al. demonstrated the benefits of combining hyperspectral imaging and ML for municipal solid waste characterization, significantly improving material identification and sorting efficiency by capturing detailed spectral information [31].

ML strategies, together with advancements in AI, have been employed in the early detection of diseases through the accurate interpretation of chest X-rays [32]. Similarly, the use of these innovations is accelerating in other areas. A valuable addition of deep learning in image recognition facilitates aircraft target recognition, enabling air defense systems to quickly determine

the target category of an acquired aircraft image and automatically estimate countermeasures, potentially saving significant reaction time and reducing combat risks [33]. In this study, we propose an automated method for identifying Conus species using a cohesive ML algorithm framework through feature-assisted training on imaging datasets. Additionally, by designing a local database, this study may serve as a basis for cataloging cone snail species, including their sequence information and family-wise distribution.

## Materials and methods

### Data collection

The image dataset of 119 *Conus* species was obtained from the ConoServer database [34]. Our proposed methodology is illustrated in the flowchart (Fig 1).

### Image preprocessing

Initially, each image file format (JPG, JPEG, or PNG) and size was checked for uniformity. The Pillow library was used to resize the images to a standard size of 224 x 224 pixels. Next, `cvtColor` was applied to find contours, and the images were converted to grayscale to remove background noise. A Canny filter was used to compute edge strength, utilizing linear filtering with a Gaussian kernel to smooth out noise [35]. The edges were then overlaid on the original RGB images. All images were processed through these steps and stored in a local folder.

We also applied some pre-processing to each highlighted image. First, using `cv2.COLOR_BGR2GRAY`, we converted the image to grayscale. Gaussian blur was applied to remove noise from each image, and the images were normalized for enhancement. We used the Canny and Sobel functions [36] with a kernel size of 5 to detect edges in each image. The original images of *Conus ammiralis*, *Conus ebraeus*, and *Conus anabathrum*, along with the binary and Canny edge-detected images, are shown in Fig 2. These species exhibit specific patterns and shapes (pointed or round). In *Conus ammiralis*, few patterns are separated by filled brown areas with varying distances, while in the case of *Conus ebraeus*, the patterns are more pronounced, making it easily distinguishable from other species. In contrast, *Conus anabathrum* contains a line pattern at the pointed end.

### Image transformation

Image transformation was performed on each pre-processed image, with the total number set to 400. We initialized the ImageDataGenerator [37] using various parameters, such as width shift range, height shift range, zoom range, and shear range, all set to 0.2. Subsequently, we modified the rotation range to 30 degrees, set the horizontal flip to 'True,' and used 'nearest' for the fill mode. Each transformed image was stored in a unique folder. For each transformation, we applied a random transformation with a size of 224 x 224 pixels. Image transformation was cross-validated before further processing. In total, we obtained 47,600 transformed images. The original *Conus andremenezi* and its transformed images are shown in Fig 3, along with a detailed description of each image, highlighting distinct height, width, and pixel count.

### Proposed methodology

The next step was to check the image quality, and all images below the standard were removed. Noisy backgrounds were eliminated, and the `cvtColor` module was used to convert the images to grayscale, followed by the application of a threshold to segment the background and obtain the largest contour. A mask was applied to remove the background. Later, we combined

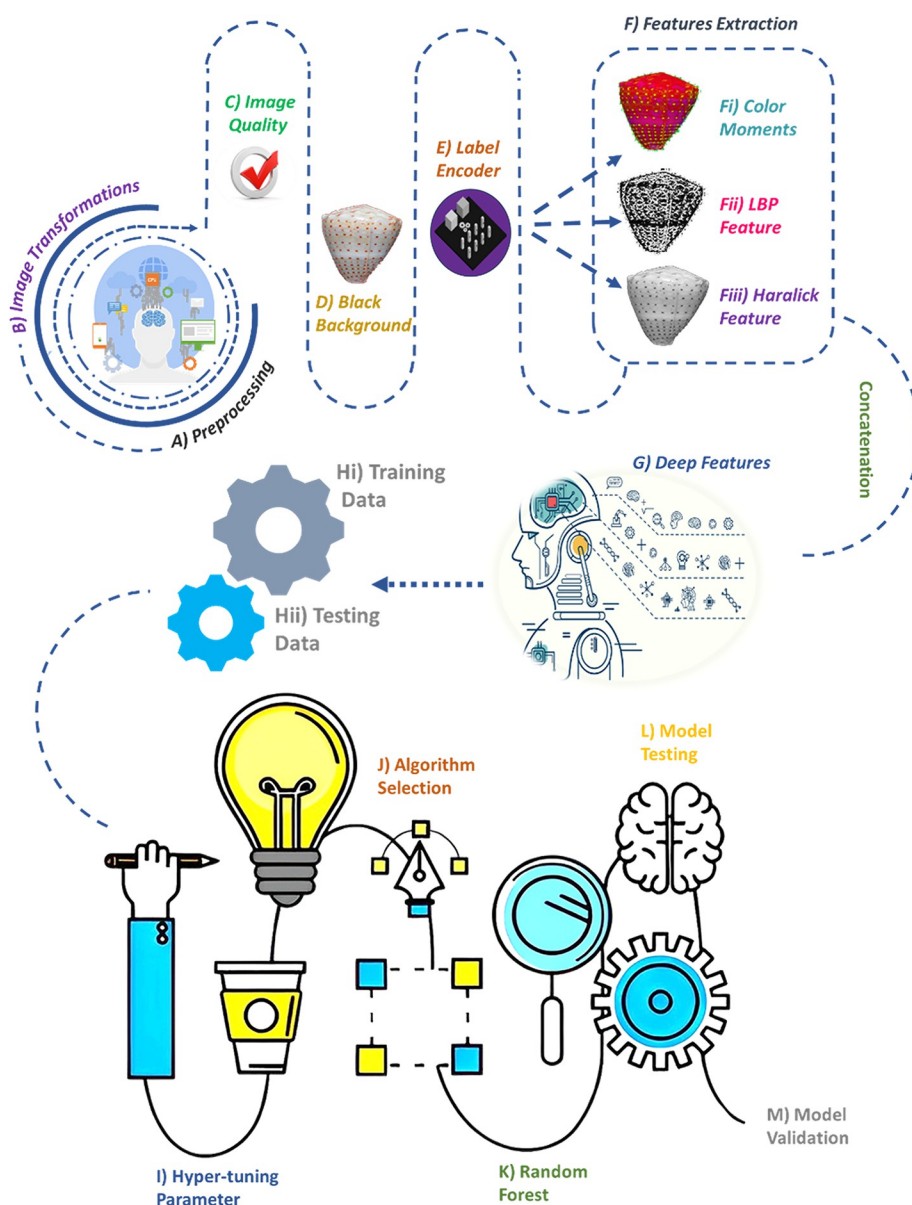

**Fig 1. Flowchart scheme of the ML-based model. A)** Image preprocessing. **B)** Image transformation. **C)** Image quality analysis of preprocessed images. **D)** Background removal by obtaining the largest contour followed by masking. **E)** Conversion of species labels into numerical values using a label encoder. **F)** Feature extraction using three different steps: **Fi)** Color moments in different orders based on color distribution. **Fii)** Texture information using local binary patterns. **Fiii)** Additional texture information using Haralick texture features. **G)** Deep feature extraction using VGG16. **H)** Training data comprising 80% of the dataset. **I)** Testing data consisting of 20% of the entire dataset. **J)** Optimization of hyperparameter tuning. **K)** Algorithm selection from all models. **L)** Random forest selection. **M)** Model testing. **N)** Model validation using different methods.

all these images into a list and used a label encoder to encode each cone snail species label as a numerical value.

**Color moments and local binary patterning.** Subsequently, color moments of different orders were calculated for each channel, revealing color distribution and variation. The local binary pattern (LBP) texture feature was computed for each grayscale image to extract texture information. LBP works by measuring the intensity levels of neighboring and central pixels,

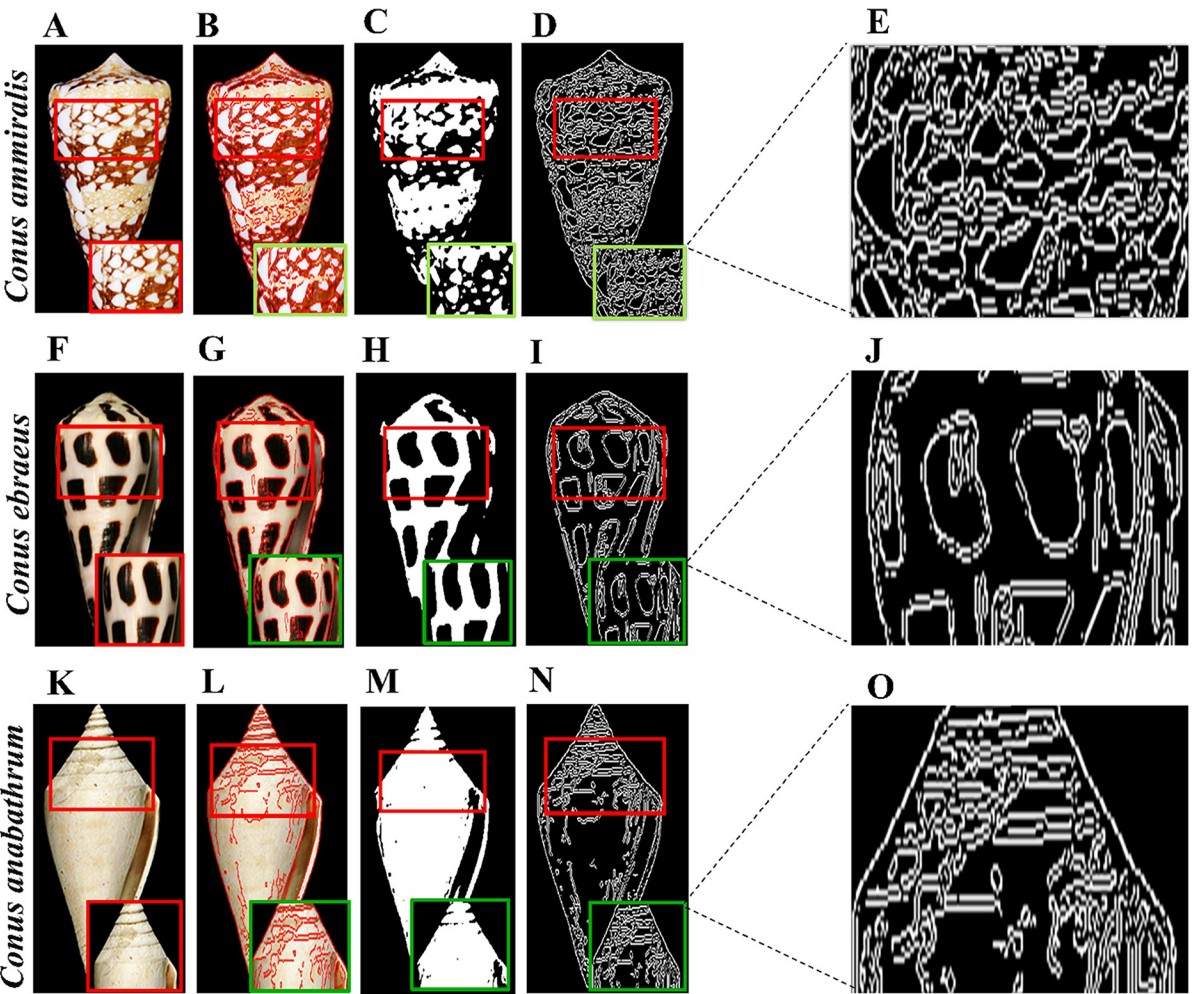

**Fig 2. Image preprocessing. A)** Original image of the *Conus ammiralis* shell, **B)** Highlighted enhanced image, **C)** Binary image, **D)** Canny edge-highlighted image, **E)** Enhanced edge-highlighted image. **F-J)** *Conus ebraeus* with enhanced, highlighted, binary, edge detected and respective enhanced images, respectively. **K-O)** *Conus anabathrum* with all respective images.

forming a binary number [38]. The threshold is obtained by comparing the neighborhood pixel $g_p$ with the center pixel $g_c$. This operator yields a binary value of 1 if $g_p$ is larger than $g_c$ and 0 otherwise. The final form of the LBP is represented in decimal value. The features extracted by the LBP operator are displayed in a histogram. This operation can be expressed as:

$$LBP_{PR} = \sum_{p=0}^{p-1} s\left(g_p - g_c\right)2^p, s(x) = \left\{ \left( \frac{1, x \geq 0}{0, x < 0} \right) \right\} \tag{1}$$

After the thresholding stage, a histogram was developed on the LBP values. With a neighborhood of P = 24P = 24P = 24 and R = 3R = 3R = 3, a 256-bin histogram represents the image features. The mathematical representation of the LBP histogram is denoted by [39]:

$$H(k) = \sum_{i=1}^{I} \sum_{j=1}^{J} f(LBP_{PR}(i,j)k), k \in [0,K], where\ f(x) = \left\{ \frac{1, x = y}{0, otherwise} \right\} \tag{2}$$

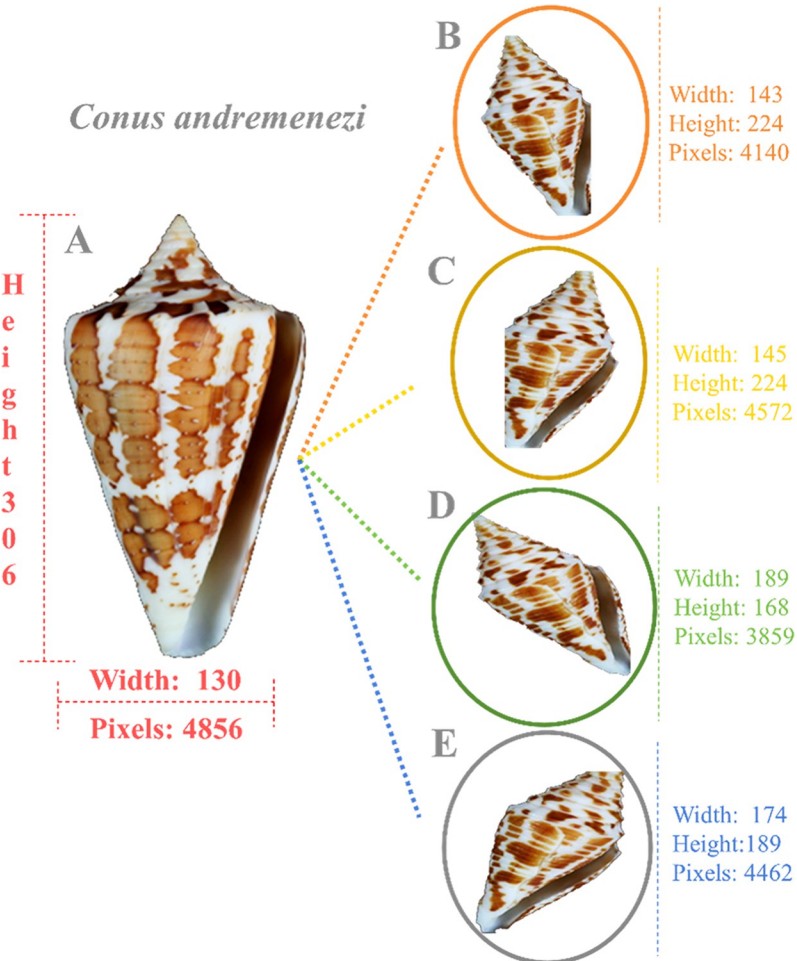

**Fig 3. Image transformation. A)** The original image of the *Conus andremenezi* shell and its dimension details are indicated in pink color. B-E) Its four transformed images, with pixel sizes ranging from 3,859 to 4,462, have different shell sizes (width × height). Each transformed image and its details are mentioned in their respective colors.

**Haralick texture feature extraction.** Next, we computed feature extraction through a method proposed by Haralick, named as spatial gray-level dependence method (SGLDM). These features are routinely used for diagnosis purposes and Alzheimer's disease diagnosis by MR images [40]. For quantifying the texture through SGLDM, 13 features were calculated in each phase. These features were extracted from the co-occurrence matrix, which represents an estimate of the second-order probability function C (i; j| x; y). This matrix represented the occurrence rate of a pixel pair with gray levels i and j, given the distances between the pixels were x and y in the x and y directions, respectively [41]. The elements of the matrix were calculated by:

$$C\left(i,j|\Delta_x\Delta_y\right) = \frac{No.of\,(x,y)\,for\,which\,I(x,y) = i, I(x+\Delta_x, y+\Delta_x) = j\,and\,both(x,y)\,and(x+\Delta_x, y+\Delta_y)\,are\,within\,the\,ROI}{No.of\,(x,y)\,for\,which\,both(x,y)\,and(x+\Delta_x, y+\Delta_y)\,are\,within\,the\,ROI} \quad (3)$$

The Haralick texture features were computed using the Haralick function, which included texture information such as contrast, correlation, and entropy in the image. In the next step, we concatenated these three features as trained features.

**Visual Geometry Group with 16-layer deep model architecture.** The Visual Geometry Group with 16-layer deep model architecture (VGG16) [42] was used for extracting deep features that were utilized in a pre-trained deep learning model. It included 16 layers, comprising 13 convolutional layers and 3 fully connected layers. VGG16 employed a small 3x3 kernel (filter) on all convolutional layers with a single stride. Max pooling layers always followed the convolutional layers. The input for VGG16 was fixed at 224 x 224 three-channel images. In VGG16, the three fully connected layers exhibited different depths. The first two layers contained a similar channel size of 4096, while the last fully connected layer had a channel size of 1000, representing the number of class labels in the ImageNet dataset. The output layer was the softmax layer, which is responsible for providing the probability of the input image [43]. We added deep features to the feature vector by horizontally stacking the deep and trained features.

**Random Forest.** The RF classifier was used due to its ideal prediction capabilities, stability, and high accuracy rate compared to a single decision tree. RF is a powerful ensemble and supervised learning method, characterized by balanced bias, minimal hyperparameter input, reduced variance, and minimized risk of overfitting in both classification and regression tasks. These features make RF an invaluable tool for prediction, modeling, and data analysis across various domains. The RF algorithm performs better with larger datasets and accelerates the decision-making process through a higher number of trees [44]. RF is an extension of the Classification and Regression Tree (CART) method, employing bagging (bootstrap aggregation) and voting to determine classification results. It consists of k classification trees, and its basic idea is to convert multiple weak classifiers into one strong classifier. The number of generated bootstrap samples determines the number of trees in the model. After the bootstrap method, each tree (bootstrap sample) is formed using the following rules: If there are M input variables, the number of m predictor variables at each node satisfies m≤Mm. The variable m is chosen randomly from M. The selection of the best predictor variable from m is determined by calculating the measure of purity (Gini or entropy). The Gini index $G_{gini}$ (D) is used to decide the optimal binary cut point for each feature. $G_{gini}$ represents the uncertainty of the set D. In the classification problem, suppose there are N classes; for a given set of samples D, the $G_{gini}$ index is:

$$G_{gini}(D) = 1 - \sum_{n=1}^{N} \left( \frac{|C_n|}{D} \right)^2 \tag{4}$$

where $C_n$ is the subset of samples in $D$ that belong to the $n$th class [45]. If a sample set $D$ is divided into two parts, $D1$ and $D2$, according to the value of feature $A$,

$$D_1 = \{(x, y) \in D | A(x) = a\}, D_2 = D - D_1 \tag{5}$$

The best split on $m$ is used to separate the nodes. The amount of $m$ is kept constant during the growth of forests. Each tree is formed to the maximum extent without pruning. The final result of RF is the optimal result chosen by voting on all classification trees [45]. The best predictor variable provides more decision-making information. More tree formation and their usage in the decision-making process yield more robust result [46].

Next, data was divided into training (80%) and testing or validation data (20%), about 38,080 and 9,520 images out of 47,600, respectively. As a result, we extracted $X_{Train}$, $X_{Valid}$, $Y_{Train}$, and $Y_{Valid}$ for further optimization of hyperparameters [47]. Enhancing the RF algorithm's ability is crucial for extracting high-quality features and optimizing parameter selection. This can significantly help reduce the model's generalization error and improve the RF

algorithm's classification accuracy. We used 100 trees or estimators and a minimum sample split of 2 for splitting the internal nodes.

The model was then trained and evaluated by fitting $X_{\text{Train}}$ and $Y_{\text{Train}}$ and by predicting the model by $X_{\text{Valid}}$.

**XGBoost.**    Tree-based gradient boosting integrated model XGBoost (XGB) [48], is composed of multiple classification regression trees (CART) that acquire the residual value through the sum of target and predicted values based on the prior decision trees. Upon training of all decision trees, they reach a consensus and finally compute the prediction result through the accumulation of samples from the previous findings. Every new tree in the XGB model training phase is trained using the previously trained tree as a model, and once a decision tree has been generated, it is stripped to avoid overfitting. The XGB model trains the obtained error to minimize the overall error. The input from each tree is utilized to train the subsequent tree again to progressively minimize the prediction error and gradually drive the model's predicted value closer to reality. The prediction model for XGB can be represented as:

$$y_i = \sum_{k=1}^{K} f_k(x_i), f_k \in F \tag{6}$$

Where $x_i$ and $y_i$ are training samples. x represents the eigenvector, y represents the sample label, and $f_k(xi)$ represents the $k$th decision tree. The corresponding objective function is defined as follows [49]:

$$Obj(O) = \sum_{i=1}^{n} L(y_i, y_i') + \sum_{k=1}^{K} \Omega(f_k) \tag{7}$$

The objective function Obj($O$) is divided into two parts: the regularization term, which reduces the chances of a model demonstrating overfitting, and the loss function, which indicates a specific objective to evaluate the accuracy of the model's prediction. The function is as follows:

$$\Omega(f) = \gamma T + \frac{1}{2}\lambda ||\omega||^2 \tag{8}$$

Where $\gamma$ is the leaf node coefficient, its goal is to optimize and modify the objective function using XGBoost, similar to a pre-pruning operation (i.e., $\gamma T$ regulates the tree's complexity; the higher the value, the higher the objective function value, which subsequently suppresses the model's complexity). The leaf node weight percentage is regulated by the full L2 regularization term, and $\lambda$, the coefficient of the squared mode of L2, prevents overfitting. The objective function is gradient boosting decision tree (GBDT) if the regularization term has a value of 0 [50].

This model lessens the chance of overfitting by including regularization elements in the objective function. It utilizes both the first and second derivatives to enhance the accuracy of the loss function and customize the loss. We used the 'Extreme Gradient Boosting' classifier of the XGB library by specifying the evaluation metric to measure cross-entropy loss (which is a multi-class logarithmic loss) and avoid any deprecation issues in the disabled labels.

## Confusion matrix

The performance of the chosen strategy was determined by a confusion matrix, which showed the number of correct and incorrect predictions made by the model as compared to the actual data [51, 52]. The confusion matrix comprises four components: True Positive (TP), True Negative (TN), False Positive (FP), and False Negative (FN). The following metrics evaluate the performance of a classification model on a dataset:

Precision = TP / (TP + FP)

Recall (Sensitivity) = TP / (TP + FN)

F1-score = 2 * (Precision * Recall) / (Precision + Recall)

Other analyses, including bar plots and histogram generation, were performed to check the proportion and prediction results through the classification report of the desired RF model. The Area Under the Receiver Operating Characteristic Curve (AUC-ROC) [53] is a performance metric for binary classification problems. The AUC-ROC value ranges from 0 to 1, where a higher value indicates better performance. A curve closer to the top-left corner represents a better model. It was plotted to estimate the true positive rate (sensitivity) against the false positive rate (specificity) at various threshold settings.

## Results

### Cone snail shell image processing

*Conus* species exhibit diverse characteristics in terms of shell shape, size, color, and localization. The differentiation characteristics, including mean intensity, intensity standard deviation, edge pixel number, mean key point, vary significantly among Conus species (Table 1). In particular, images obtained from different sources need to be processed for color variation, background noise removal, pixel adjustment, and color intensity correction. To accurately process shell images, we scaled the RGB (red, green, and blue) intensity in the image. The average predicted RGB values were 70.23, 88.12, and 107.98 for R, G and B, respectively (S1 Fig). These values were distinct for each image, which largely facilitated enhancing model efficiency.

The dataset of 47,600 images were split into 80% training and 20% testing data, resulting in 38,080 and 9,520 images. $X_{Train}$, $X_{\text{Valid}}$, $Y_{\text{Train}}$, and $Y_{\text{Valid}}$ were extracted for hyperparameter optimization [54]. Enhancing the RF algorithm is crucial for extracting high-quality features and optimizing parameter selection. This may significantly help reduce the model's generalization error and improve the RF algorithm's classification accuracy. The model was trained and evaluated by fitting $X_{Train}$ and $Y_{Train}$ and by predicting with $X_{\text{Valid}}$.

### Model validation

Next, we added more data to check the predictions for each search image as validation data. Among the 119 species, five species were wrongly predicted: *Conus monile* was predicted as *Conus kintoki*, *Conus monachus* was predicted instead of *Conus virgo*, *Conus **tinianus*** as *Conus catus*, *Conus vitulinaus* was predicted as *Conus regularis* and *Conus flavidus* was predicted as *Conus betulinus*. All other species were accurately predicted by the trained RF model, achieving a high accuracy rate (S1 Table). For these species, structural similarity index ranged from 0.33 to 0.99, which measures similarity between test and reference images by calculating variations in contrast, brightness, and edge information [55].

We included images of some species other than cone snails, such as *Miter shells*, *Olive shells*, *Cypraea argus*, *Aulica imperialis*, and *Eloise Beach*, along with Conus species *Conus literatus*, *Conus asiaticus*, and *Conus ebraeus* for further validation of our model (Fig 4). Training results revealed no irrelevant species due to feature differentiation. These shell images were ranked in the range of 27,674, 27,413, 27,584, 26,522, and 26,549, while Conus shells exhibited 27,143 features. Overall, the proposed model in this report is 95% efficient in cone snail species recognition through shell images.

### Model performance assessment

**Precision and recall analysis.** The RF classification report indicated a significant proportion of TP predictions as compared to XGB. Multiple species exhibiting precision score values

**Table 1. Statistical analysis of raw images of Conus species before preprocessing.** Size (S), mean intensity (MI), intensity standard deviation (ISD), number of edge pixels (NEP), and mean key point size (MKS) are presented in different columns.

| Specie name | Size | MI | ISD | NEP | MKS |
|---|---|---|---|---|---|
| *Conus abbreviatus* | 126 x 196 | 114.900551 | 85.24146037 | 2070 | 3.855107131 |
| *Conus achatinus* | 234 x 469 | 90.1004501 | 72.62897095 | 17287 | 3.738486035 |
| *Conus adamsonii* | 166 x 309 | 81.5929543 | 64.52867715 | 10219 | 3.394759074 |
| *Conus amadis* | 137 x 283 | 89.7953625 | 81.18837988 | 7880 | 3.242136133 |
| *Conus ammiralis* | 147 x 266 | 101.342642 | 90.00906917 | 7293 | 3.558139329 |
| *Conus anabathrum* | 113 x 236 | 102.825859 | 91.64173238 | 2811 | 5.372164498 |
| *Conus andremenezi* | 130 x 306 | 77.2646053 | 78.52747464 | 4856 | 4.783157641 |
| *Conus anemone* | 140 x 333 | 102.90532 | 87.34408651 | 6528 | 5.295740278 |
| *Conus araneosus* | 190 x 344 | 95.4301561 | 89.64749384 | 10002 | 4.042105765 |
| *Conus archon* | 173 x 325 | 78.0636372 | 76.09148881 | 6231 | 3.903034503 |
| *Conus arenatus* | 150 x 258 | 127.868966 | 88.99116882 | 5384 | 3.248078797 |
| *Conus aristophanes* | 125 x 209 | 115.918813 | 84.83766222 | 3292 | 3.536915887 |
| *Conus asiaticus* | 160 x 303 | 91.0680693 | 93.87201536 | 5579 | 3.533550901 |
| *Conus ateralbus* | 147 x 251 | 65.5442177 | 66.74975933 | 7071 | 3.789925593 |
| *Conus aulicus* | 127 x 305 | 95.9198916 | 75.00209789 | 6079 | 3.877136884 |
| *Conus aurisiacus* | 172 x 309 | 99.8640777 | 80.66912726 | 5803 | 3.930534717 |
| *Conus austini* | 167 x 318 | 82.4746545 | 77.49197812 | 4362 | 3.392523493 |
| *Conus australis* | 115 x 306 | 94.9256323 | 82.35290923 | 5929 | 3.427619775 |
| *Conus bandanus* | 646 x 1202 | 83.4092946 | 78.0616978 | 44765 | 7.77255379 |
| *Conus bayani* | 114 x 227 | 67.7127676 | 71.5729914 | 2925 | 4.299335957 |
| *Conus betulinus* | 224 x 335 | 101.927159 | 81.59713297 | 7407 | 3.679199442 |
| *Conus brunneus* | 154 x 191 | 66.2547766 | 62.9423698 | 6141 | 3.567691536 |
| *Conus bullatus* | 114 x 219 | 107.642193 | 67.27182937 | 5300 | 3.453630916 |
| *Conus californicus* | 462 x 846 | 80.9327341 | 69.53473463 | 15680 | 5.947179261 |
| *Conus capitaneus* | 169 x 252 | 80.4169954 | 65.4770939 | 7079 | 3.501728312 |
| *Conus caracteristicus* | 163 x 225 | 102.626667 | 82.49999629 | 5106 | 3.704006016 |
| *Conus catus* | 135 x 240 | 90.7333025 | 71.40495938 | 6002 | 3.917673782 |
| *Conus cervus* | 136 x 274 | 100.219381 | 76.16191665 | 6241 | 3.530832996 |
| *Conus chiangi* | 153 x 264 | 85.9632601 | 76.05567255 | 5952 | 3.216992084 |
| *Conus circumcisus* | 116 x 279 | 109.326319 | 72.72633786 | 5200 | 4.345783836 |
| *Conus consors* | 141 x 299 | 86.4395266 | 67.08034587 | 2973 | 6.904867876 |
| *Conus coronatus* | 83 x 133 | 97.6019567 | 81.97972403 | 2303 | 3.625967436 |
| *Conus dalli* | 157 x 267 | 93.0402681 | 77.21764877 | 8964 | 3.322457316 |
| *Conus delessertii* | 161 x 307 | 83.4159063 | 81.82363518 | 5977 | 5.091297852 |
| *Conus diadema* | 194 x 307 | 89.0702173 | 73.76674884 | 7057 | 4.010664793 |
| *Conus distans* | 89 x 160 | 105.306812 | 85.94422289 | 2571 | 4.76245108 |
| *Conus ebraeus* | 209 x 311 | 73.9946615 | 79.35979636 | 5273 | 5.99292686 |
| *Conus eburneus* | 222 x 349 | 92.4428612 | 88.25812038 | 8119 | 5.578738826 |
| *Conus emaciatus* | 251 x 405 | 81.1571197 | 60.17887854 | 4533 | 5.640602514 |
| *Conus episcopatus* | 150 x 320 | 91.8544583 | 80.46877076 | 9376 | 3.610899895 |
| *Conus ermineus* | 185 x 329 | 88.6190421 | 76.6154946 | 6962 | 4.545134057 |
| *Conus ferrugineus* | 210 x 416 | 83.9429831 | 70.89388376 | 7075 | 5.904191236 |
| *Conus figulinus* | 282 x 407 | 80.4968547 | 71.85676559 | 16639 | 3.399929217 |
| *Conus flavidus* | 170 x 295 | 97.7092921 | 74.24410961 | 4067 | 3.978458209 |
| *Conus floridulus* | 667 x 1131 | 87.5026585 | 80.04923969 | 14478 | 7.446896809 |
| *Conus frigidus* | 156 x 265 | 103.462821 | 72.74093208 | 4527 | 3.876202816 |

*(Continued)*

**Table 1.** (Continued)

| Specie name | Size | MI | ISD | NEP | MKS |
|---|---|---|---|---|---|
| *Conus fulmen* | 196 x 357 | 83.4306723 | 72.73174627 | 2611 | 6.047156509 |
| *Conus gauguini* | 89 x 163 | 103.28214 | 76.99108538 | 2067 | 4.176049745 |
| *Conus generalis* | 135 x 287 | 98.3426249 | 85.07910231 | 2625 | 4.553597675 |
| *Conus geographus* | 76 x 178 | 69.1569338 | 59.76345165 | 3261 | 4.091072835 |
| *Conus gladiator* | 168 x 249 | 82.6303548 | 72.62570646 | 5498 | 4.632142848 |
| *Conus gloriamaris* | 119 x 343 | 80.1448661 | 70.72475205 | 10025 | 3.040092381 |
| *Conus imperialis* | 82 x 156 | 73.2795497 | 77.08575888 | 3229 | 3.322255486 |
| *Conus inscriptus* | 161 x 330 | 102.096951 | 88.50804845 | 6676 | 4.308585652 |
| *Conus judaeus* | 186 x 311 | 80.9460291 | 91.12290436 | 5677 | 5.282902826 |
| *Conus kinoshitai* | 133 x 306 | 109.052312 | 87.59792953 | 4673 | 4.002651231 |
| *Conus kintoki* | 171 x 362 | 110.7324 | 82.19116586 | 3015 | 3.805836274 |
| *Conus leopardus* | 120 x 211 | 100.795616 | 79.70195154 | 5007 | 3.217252134 |
| *Conus limpusi* | 166 x 335 | 80.9515015 | 66.41763157 | 2640 | 5.178752613 |
| *Conus litteratus* | 91 x 156 | 128.705128 | 101.7086228 | 2609 | 2.857512904 |
| *Conus lividus* | 137 x 249 | 88.3807639 | 77.07639339 | 2551 | 4.272186609 |
| *Conus longurionis* | 116 x 351 | 79.9143089 | 72.36931846 | 5730 | 4.66598781 |
| *Conus loroisii* | 172 x 273 | 54.6943948 | 46.1643957 | 9412 | 3.114201716 |
| *Conus lynceus* | 174 x 386 | 106.663123 | 82.14609781 | 8848 | 5.149299075 |
| *Conus magnificus* | 116 x 261 | 111.673438 | 84.23736161 | 7044 | 3.127516587 |
| *Conus magus* | 279 x 582 | 102.056319 | 77.61301005 | 21916 | 5.131758487 |
| *Conus marmoreus* | 464 x 987 | 71.0377253 | 75.10760805 | 35820 | 8.258521537 |
| *Conus memiae* | 210 x 350 | 78.1916871 | 85.53722763 | 9499 | 5.492338902 |
| *Conus miles* | 136 x 207 | 74.5656081 | 76.81322467 | 5401 | 3.088816641 |
| *Conus miliaris* | 180 x 296 | 90.4191254 | 74.03838125 | 8054 | 3.592617067 |
| *Conus milneedwardsi* | 69 x 223 | 87.232274 | 80.6165129 | 2860 | 3.355763269 |
| *Conus monachus* | 226 x 424 | 118.874885 | 87.78246128 | 11436 | 4.074395915 |
| *Conus moncuri* | 195 x 342 | 83.8184885 | 75.0833367 | 7482 | 4.94113918 |
| *Conus monile* | 161 x 337 | 87.6115709 | 82.27396067 | 5175 | 4.915662615 |
| *Conus mus* | 84 x 150 | 95.6694444 | 76.5681594 | 3144 | 3.750667921 |
| *Conus mustelinus* | 149 x 272 | 92.0070322 | 77.31240476 | 5208 | 4.039098181 |
| *Conus natalis* | 157 x 318 | 74.0708449 | 67.42437807 | 10205 | 4.775390739 |
| *Conus nigropunctatus* | 127 x 216 | 92.907699 | 73.07321996 | 4902 | 4.230405607 |
| *Conus nux* | 194 x 332 | 77.2799186 | 71.9322076 | 4320 | 7.542459114 |
| *Conus obscurus* | 70 x 160 | 74.8146429 | 52.24022396 | 2667 | 3.338338166 |
| *Conus omaria* | 86 x 194 | 107.138576 | 65.92997092 | 4084 | 3.085670003 |
| *Conus parius* | 182 x 303 | 100.761923 | 82.87203435 | 1756 | 4.551220399 |
| *Conus pennaceus* | 104 x 174 | 78.9077697 | 86.20071699 | 2587 | 4.316248887 |
| *Conus pergrandis* | 136 x 344 | 76.2034456 | 77.06110854 | 5116 | 3.972704224 |
| *Conus pictus* | 182 x 340 | 77.9745637 | 73.67581946 | 7533 | 4.965593014 |
| *Conus planorbis* | 113 x 207 | 75.9901672 | 69.28186747 | 4616 | 3.446087527 |
| *Conus princeps* | 155 x 273 | 110.268746 | 91.49685066 | 5370 | 4.075615161 |
| *Conus profundineocaledonicus* | 155 x 333 | 87.7985857 | 74.28470619 | 1865 | 6.221098957 |
| *Conus purpurascens* | 554 x 932 | 64.4792845 | 59.42204983 | 57556 | 4.533820502 |
| *Conus quercinus* | 160 x 272 | 106.95347 | 78.62236341 | 1603 | 10.07662979 |
| *Conus radiatus* | 114 x 244 | 81.7528401 | 61.54902861 | 2578 | 3.766189418 |
| *Conus rattus* | 185 x 298 | 85.1469617 | 71.41448917 | 7632 | 4.262023336 |
| *Conus regius* | 146 x 261 | 89.9245263 | 76.46770287 | 7479 | 3.76844333 |

*(Continued)*

**Table 1.** (Continued)

| Specie name | Size | MI | ISD | NEP | MKS |
|---|---|---|---|---|---|
| *Conus regularis* | 134 x 285 | 86.8535219 | 78.17037418 | 5597 | 4.388107317 |
| *Conus rolani* | 151 x 300 | 106.889382 | 82.66212069 | 4221 | 4.337546096 |
| *Conus sanguinolentus* | 153 x 262 | 91.726987 | 73.29265179 | 2704 | 5.807804724 |
| *Conus sponsalis* | 304 x 381 | 85.6163835 | 83.54710968 | 7368 | 5.747592142 |
| *Conus spulicarius* | 216 x 346 | 86.9485389 | 74.4378499 | 9807 | 5.313243719 |
| *Conus spurius* | 166 x 270 | 106.758188 | 82.90364202 | 3524 | 5.422410713 |
| *Conus stercusmuscarum* | 113 x 236 | 111.163154 | 77.06938388 | 4015 | 3.110592977 |
| *Conus striatus* | 135 x 306 | 109.730864 | 80.91233496 | 6460 | 4.263692126 |
| *Conus striolatus* | 149 x 268 | 90.6919764 | 74.36637035 | 7842 | 4.19279689 |
| *Conus sulcatus* | 150 x 266 | 87.610802 | 73.93412992 | 7816 | 3.80518956 |
| *Conus sulturatus* | 109 x 175 | 123.898768 | 81.81499109 | 735 | 10.90305368 |
| *Conus terebra* | 102 x 237 | 104.010176 | 80.97356016 | 1960 | 4.921096532 |
| *Conus tessulatus* | 163 x 252 | 86.5140715 | 76.07680221 | 4052 | 5.229228191 |
| *Conus textile* | 114 x 228 | 88.8001693 | 75.83345613 | 6716 | 2.816846265 |
| *Conus tinianus* | 99 x 192 | 104.217119 | 77.25270464 | 2544 | 4.685287444 |
| *Conus tulipa* | 115 x 228 | 105.702021 | 65.36394178 | 6445 | 3.410149088 |
| *Conus varius* | 136 x 266 | 104.889761 | 82.3494734 | 3056 | 6.092001697 |
| *Conus ventricosus* | 158 x 277 | 93.4467395 | 81.42046926 | 9519 | 3.198309433 |
| *Conus vexillum* | 152 x 249 | 96.2798563 | 81.22803912 | 6762 | 4.179350178 |
| *Conus victoriae* | 86 x 183 | 66.5662727 | 67.5775656 | 3900 | 3.02614837 |
| *Conus villepinii* | 76 x 183 | 94.9417601 | 87.01109306 | 2081 | 4.106920018 |
| *Conus virgo* | 164 x 316 | 109.196955 | 84.3636481 | 1928 | 4.158706044 |
| *Conus vitulinus* | 146 x 282 | 93.9529049 | 78.04338245 | 4788 | 4.153326996 |
| *Conus ximenes* | 80 x 140 | 93.75125 | 81.05532005 | 2199 | 2.958279716 |
| *Conus zeylanicus* | 146 x 251 | 125.553376 | 92.91948417 | 6446 | 3.69964845 |
| *Conus zonatus* | 66 x 129 | 94.6779422 | 74.00836486 | 2014 | 3.002186416 |

close to 1 demonstrated accurate predictions through the RF model. These species were categorized into three groups for better representation in bar plots (Fig 5). Among the 119 Cone snail species, group 1 contained 40 species, group 2 exhibited 39 species, and group 3 included 40 members.

In group 1, nine members (*Conus andremenezi*, *archon*, *aurisiacus*, *austini*, *bandanus*, *californicus*, *delessertii*, *diadema* and *episcopatus*) exhibited RF precision scores of 0.98, 0.92, 0.87, 0.95, 0.98, 0.96, 0.89, 0.96, and 0.90, respectively. Group 2 comprised 15 members (*ermineus*, *figulinus*, *floridulus*, *frigidus*, *fulmen*, *geographus*, *inscriptus*, *judaeus*, *lividus*, *magus*, *memiae*, *miles*, *miliaris*, *mustelinus* and *nux*) demonstrating precision scores of 0.95, 0.99, 0.95, 0.93, 0.94, 0.93, 0.95, 0.96, 0.96, 0.92, 0.94, 0.96, 0.96, 0.96, 0.99 scores. In contrast, 10 species in group 3, including *obscurus*, *pergrandis*, *pictus*, *planorbis*, *purpurascens*, *sponsalis*, *striolatus*, *sulcatus*, *varius*, *and ventricosus* contained precision scores of 0.97, 0.91, 0.93, 0.95, 0.99, 0.94, 0.99, 0.96, 0.90, 0.95, respectively through the RF model. The minimum precision value (0.64) was observed for *Conus consors*.

Notably, *Conus anabathrum*, *araneosus*, *kintoki*, and *sanguinolentus* exhibited better precision scores using XGB. Nevertheless, the high proportions of TP predictions among actual positive instances underscored the effectiveness of the RF model. The presence of a high recall value (a measure of model quantity) further bolstered the model's accuracy, with 24 species considered FN. *Conus lividus* exhibited a score of 0.8227. These 24 species were *ammiralis*,

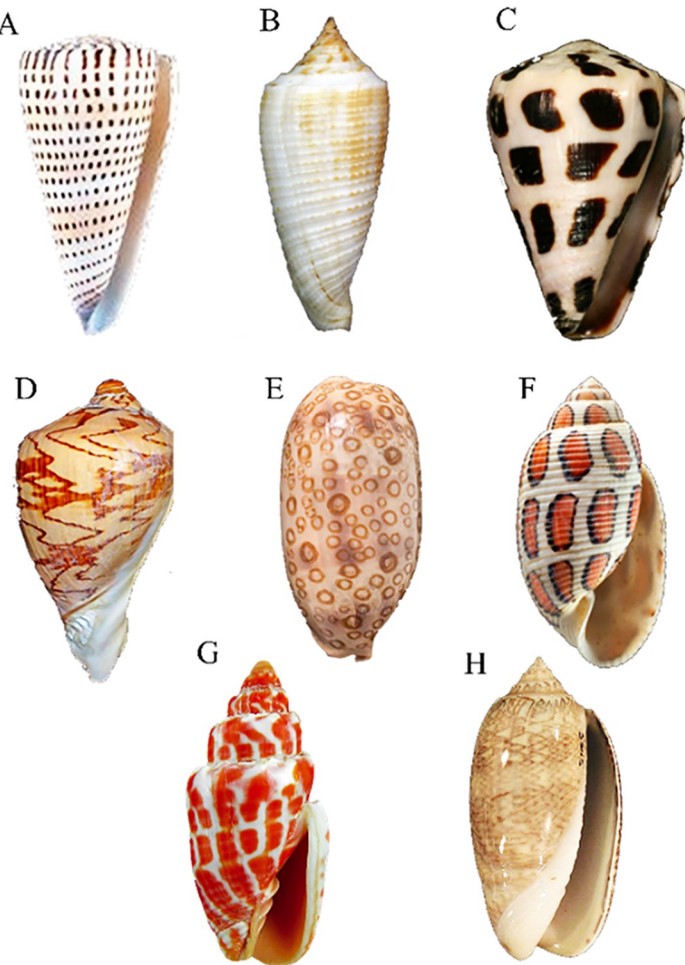

**Fig 4. Prediction results of species other than Conus species. A-C)** Conus species that are accurately recognized by our model as *Conus litteratus*, *Conus asiaticus*, and *Conus ebraeus*, respectively. **D-H)** Feature differentiation led to no species recognition in cases of *Aulica imperalis*, *Cypraea argus*, *Eloise beach*, *Miter shells*, and *Olive shells*.

*anabathrum, australis, bandanus, californicus, coronatus, dalli, episcopatus, fulmen, gloria-maris, imperialis, litteratus, loroisii, lynceus, marmoreus, miliaris, milneedwardsi, natalis, obscurus, parius, rattus, striolatus, sulcatus, zeylanicus.* Out of these, 7 species were members of group 1, 11 were in group2, and 6 species were part of group 3. The recall scores for the XGB model ranged from 0.80–0.98 (Fig 5). The harmonic mean of precision and recall, known as the F1 score, ranged from 0.76 to 1 for the RF model. It balances precision and recall, serving as a single metric for evaluating model performance. The number of actual occurrences of each class in the dataset was captured by the support value. We focused on the RF model for further validation and evaluation results.

**F1 score and support analysis.**    The F1 score (harmonic mean) ranged from 0.76 to 1 for the RF model, revealing a balanced performance between recall and precision. The class distribution was analyzed by examining the support, reflecting actual class occurrences. The F1 score and support plots demonstrated model performance across several classes. The model accurately predicted multiple classes with high F1 scores. *Conus sanguinolentus* was observed in the range of 0.82 to 0.83, while other species fell within the ranges of 0.85–0.88, 0.88–0.91, 0.91–0.94, 0.94–0.97, and 0.97–0.99, with counts of 6, 9, 17, 34, and 43 species, respectively.

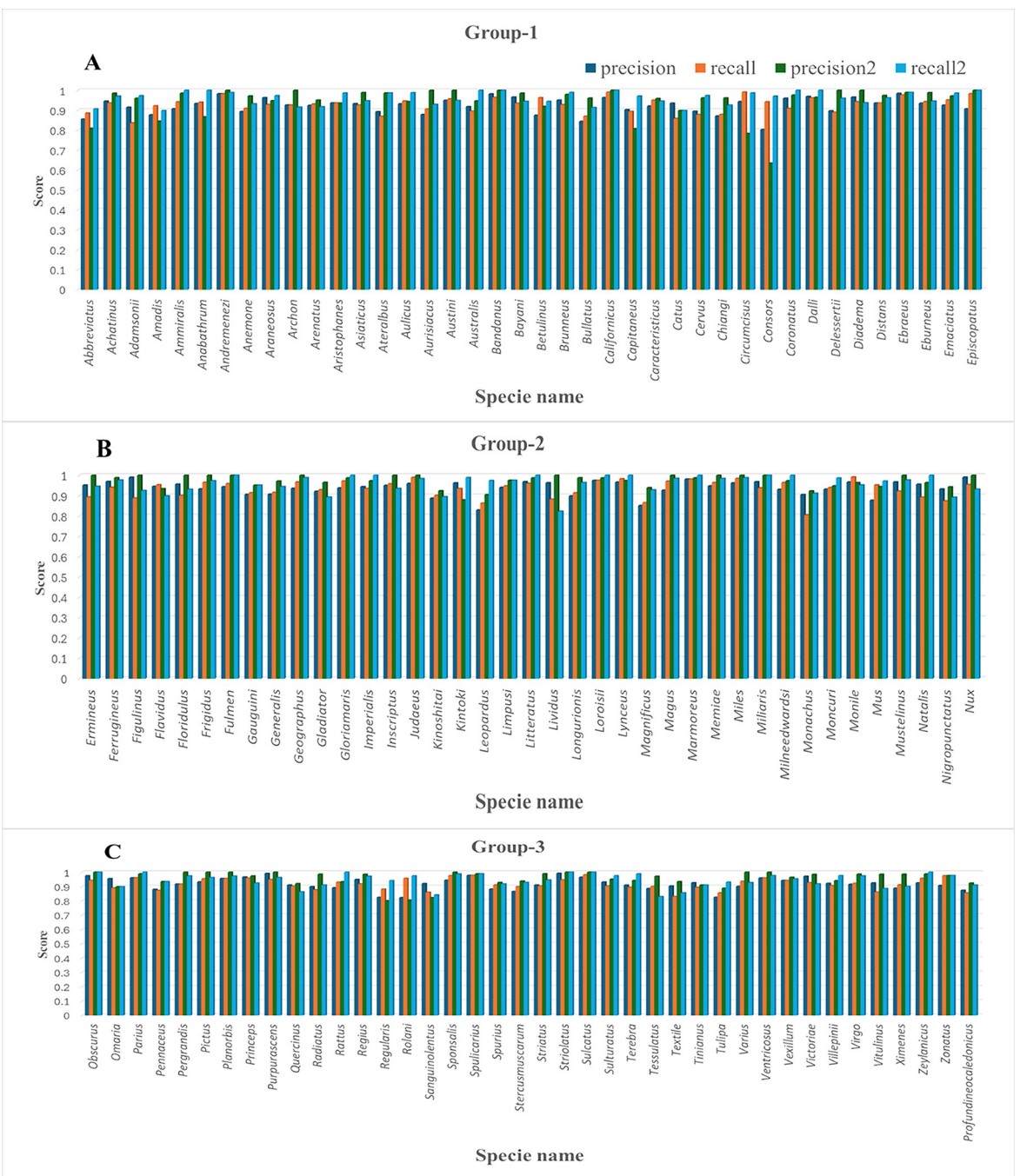

**Fig 5. Bar plot for precision and recall values for 119 Cone snail species are categorized into three groups.** Bar plot illustrating precision and recall values for 119 Cone snail species categorized into three groups. **A)** Group 1 contains 40 species. **B)** Group 2 exhibits 39 species, while C) Group 3 comprises 40 members. In all plots, species names are presented on the X-axis, while the corresponding precision and recall rates obtained through RF and XGB models are indicated on the Y-axis. The dark blue and orange bars represent the respective values of precision and recall for each species by XGB, while the green and blue bars represent precision and recall values obtained by the RF model.

Eight species exhibited maximum scores, including *Conus bandanus*, *californicus*, *episcopatus*, and *fulmen* from group 1 (Fig 6A), while *miliaris*, *obscurus*, *striolatus*, and *sulcatus* belonged to group 2 (Fig 6B). Some classes with low F1 scores were also observed, such as *Conus consors*

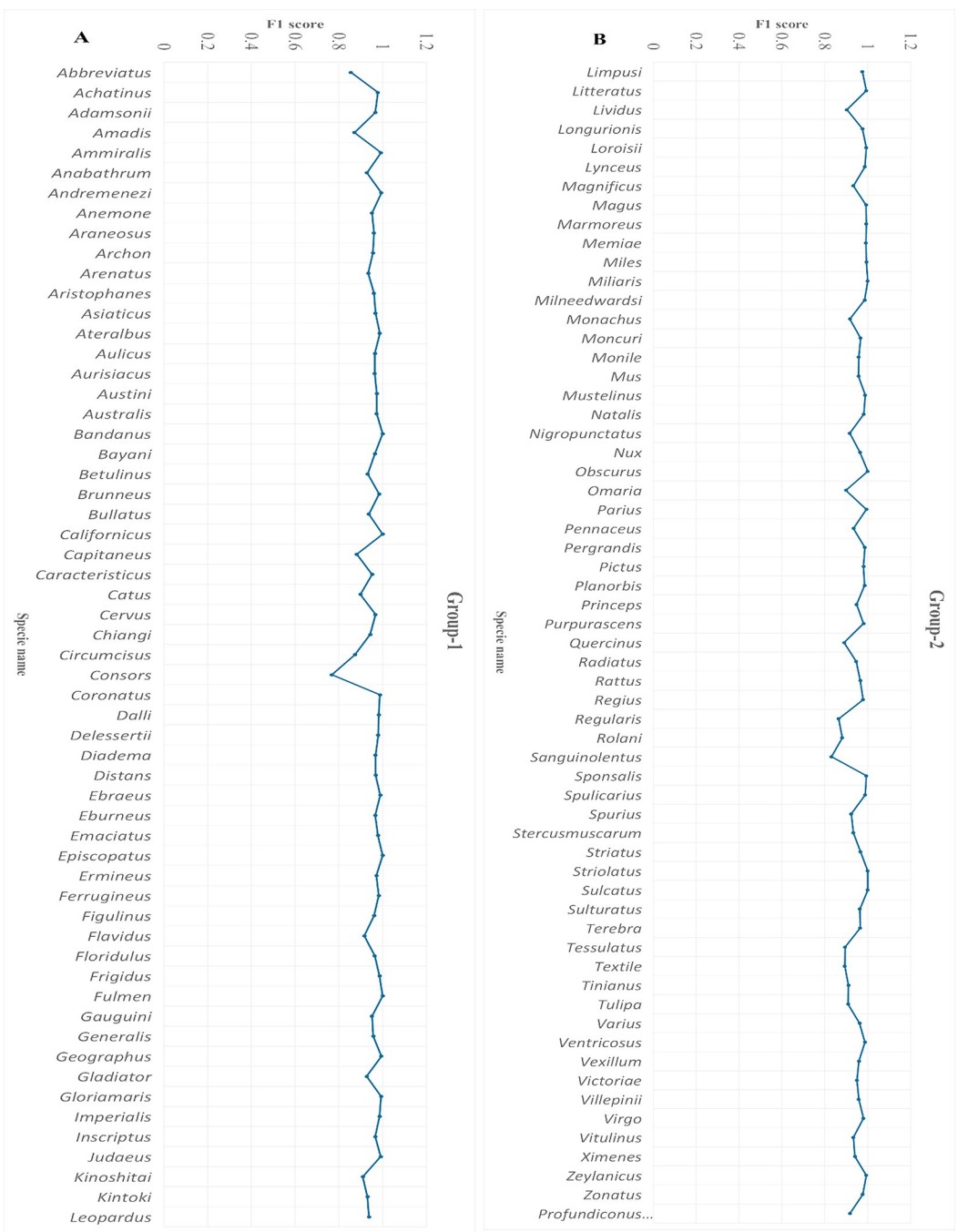

**Fig 6. F1 score analysis.** The line graphs indicate the performance scores for each class in the dataset. A) Group 1 contains 59 species (X-axis) against performance scores (Y-axis). B) Group 2 comprises 60 species (X-axis) with their respective F1 scores (Y-axis). The blue color indicates the F1-score values, showcasing the model's accurate predictions for multiple classes with high F1 scores.

with a score of 0.76, indicating slightly poor prediction. Overall, these findings provided evidence that the model operated effectively with significant F1 score values.

To comprehend class distribution, a support analysis was performed. The histogram indicated varying class numbers in terms of their distribution. *Conus sulcatus* exhibited a score in

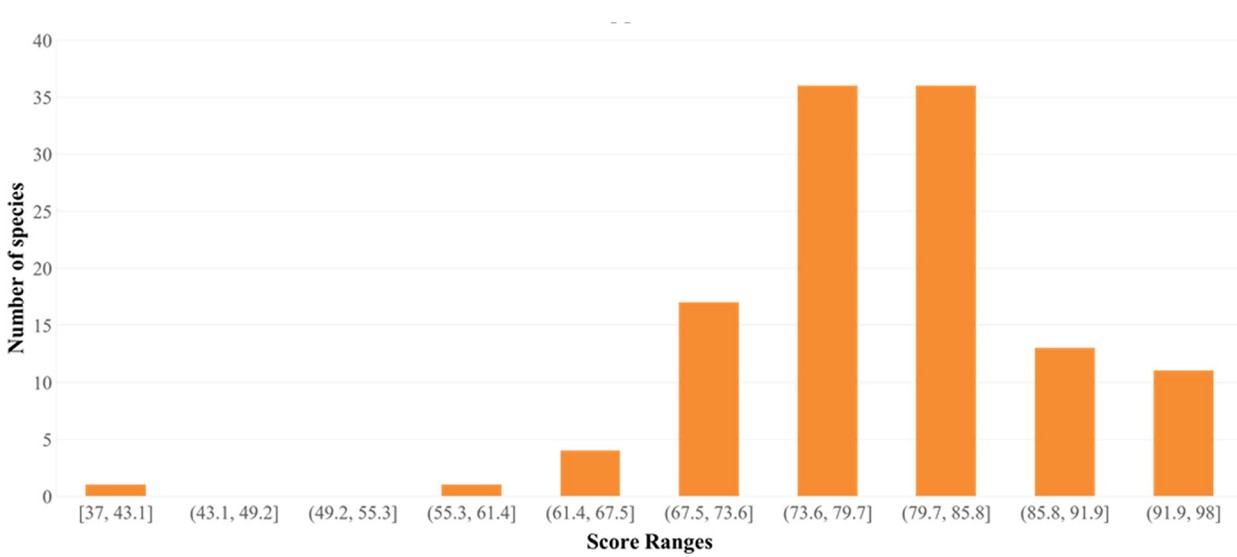

**Fig 7. Support value histogram plot.** It indicates the variation in species distribution patterns against score ranges obtained through the model classification report.

the range of 37 to 43. One, four, and seventeen species were noticed in the ranges of 55.3–61.4, 61.4–67.5, and 67.5–73.6, respectively. The number of species significantly increased to 72 for the range of 73.6–85.8. Finally, 13 and 11 species were observed with the highest range values of 85.8–91.9 and 91.9–98, respectively (Fig 7). Classes with high support values were well represented in the dataset, whereas those with low values were less common.

## Confusion matrix

A confusion matrix revealed the instances where the RF model accurately predicted a positive class. The FPR is represented by a negative value. Confusion matrix analysis revealed 24 species with TPR values of 1, indicating accurate predictions. These species included *Conus ammiralis, anabathrum, australis, bandanus, californicus, coronatus, dalli, episcopatus, fulmen, gloriamaris, imperialis, litteratus, loroisii, lynceus, marmoreus, miliaris, milneedwardsi, natalis, obscurus, parius, rattus, striolatus, sulcatus, and zeylanicus.* The lowest TPR values were 0.8227 and 0.8292 for *Conus tessulatus* and *Conus lividus*, respectively. For FNR, values should be close to zero, indicating instances where the model incorrectly predicts a negative class as positive, while TNR denotes the correct prediction of the negative class. FNR values for all 24 species were zero. In contrast, *Conus lividus* and *Conus tessulatus* exhibited the highest FNR values of 0.177 and 0.171, respectively.

A deeper insight into the model's performance was obtained using a heatmap. Fig 8 represents the macro average, average, and weighted average of recall, precision, and F1 scores based on the values obtained from the model. Due to the narrow range (0.955–0.958), color differences were minimal. Darker hues (purple) indicated somewhat lower values (0.955) for accuracy in F1 score, recall, precision, and weighted average of recall. In contrast, lighter hues indicated slightly higher values. These findings suggest that all metrics and classes contributed to consistent model performance. The highest weighted precision average was 0.958, indicating improved performance.

## Model performance evaluation

To evaluate model performance, both training and validation scores were plotted (Fig 9A and 9B). The validation curve showed a high training score across the range of hyperparameters, suggesting that the model fit the training data very well. The validation score curve indicated

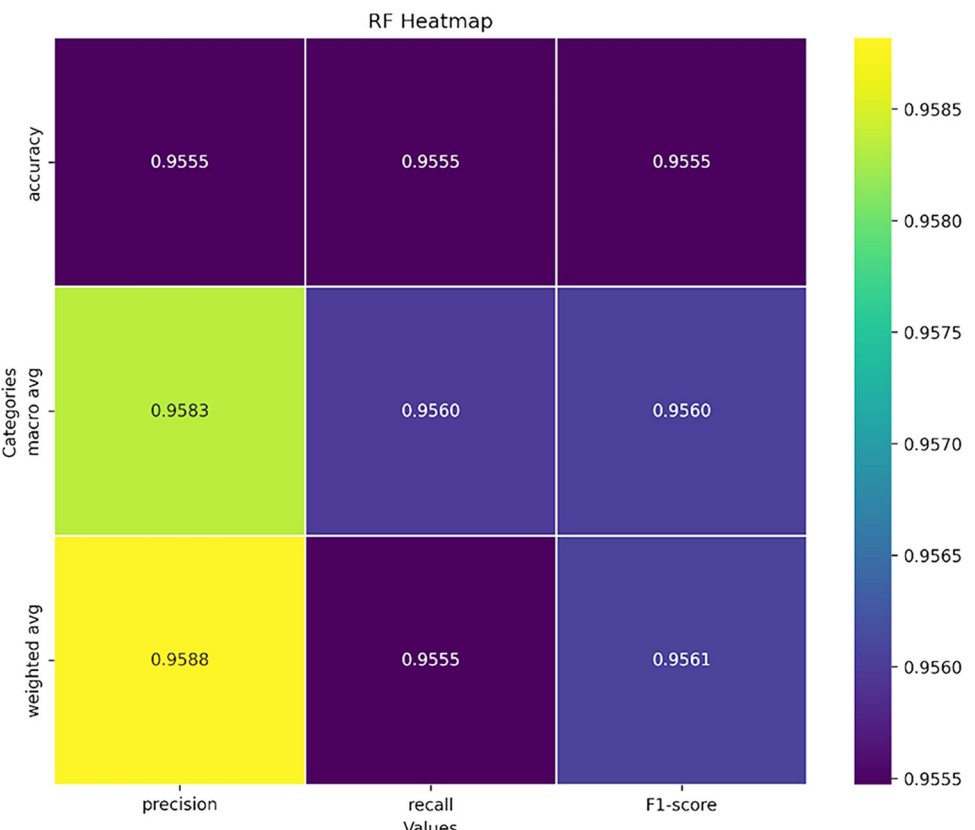

**Fig 8. Heatmap of different categories against the precision, recall, and F1-score.** Categories include accuracy, macro-average, and weighted average. The color variations from darker to lighter indicate differences in their values.

that the model generalized well to unseen data for these hyperparameter values. Both training and validation scores were high and closely aligned, reflecting a good balance between bias and variance. This indicates that the model is well-performing and appropriately tuned, with strong generalization capabilities (Fig 9A).

In the learning curve, a training score close to 1 (or 100%) revealed that the model learned and fitted the training data effectively. The validation score stabilized at approximately 95%, indicating good generalization performance for new data. The small gap between training and validation scores suggests that the model's complexity is appropriate for the given data, achieving a favorable balance between variance and bias (Fig 9B). The model is neither significantly overfitted, as it performs well on both training and validation datasets, nor underfitted (as both training and validation scores are low), making it a "Good Fit" model.

Next, we plotted a Precision-Recall (PR) curve, which shows precision against recall for different thresholds. A curve closer to the top-right corner indicates better model performance. The area under the PR curve serves as a single metric to assess overall performance (Fig 9C). Thus, the current model demonstrates favorable precision and recall values, indicating its accurate prediction ability.

## Discussion

Identifying *Conus* species presents significant challenges due to the similarities in shell patterns among various mollusks. The classification of cone snail taxonomic features requires

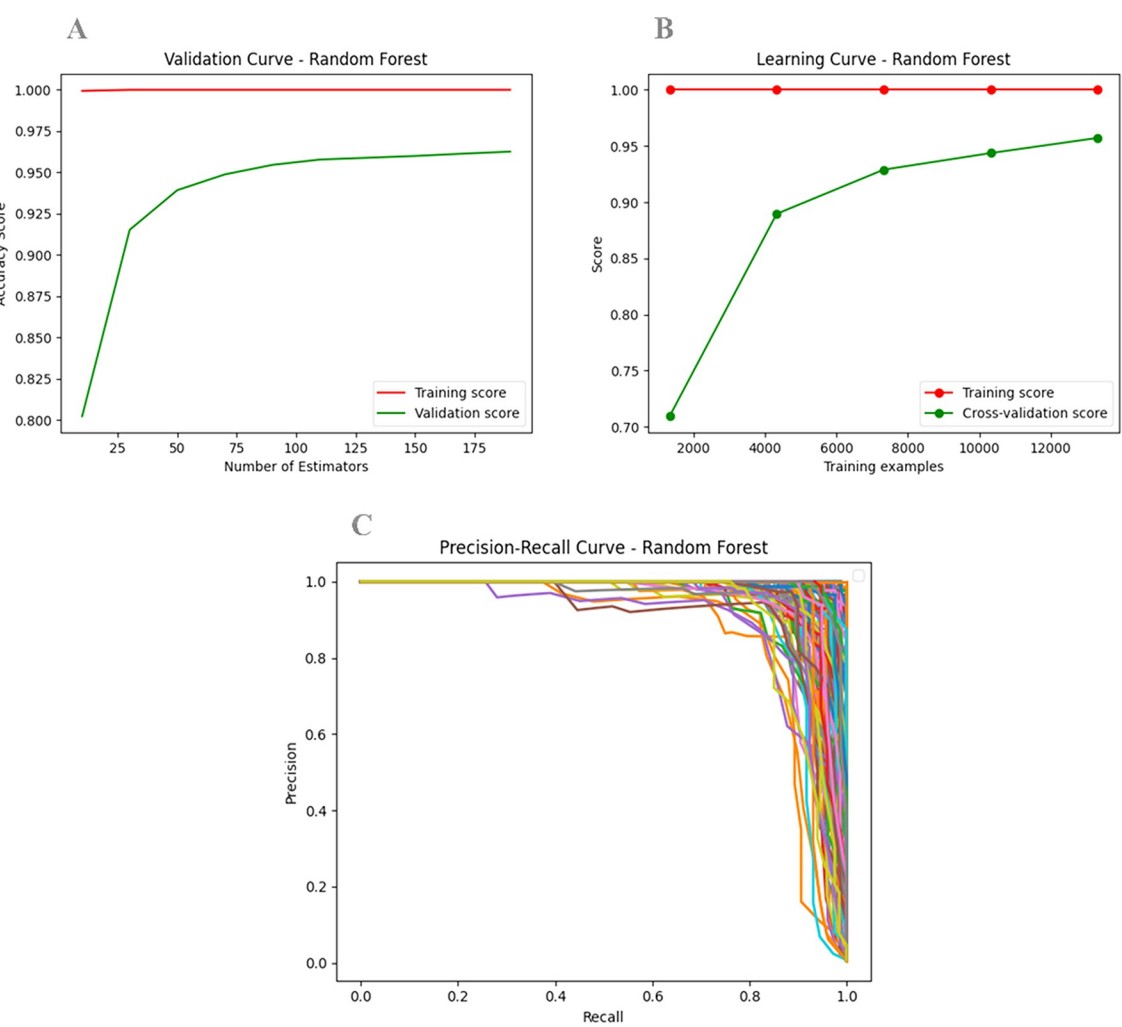

**Fig 9. Model performance analysis. A)** The validation curve plots hyperparameter values (X-axis) against model performance metrics (accuracy score on the Y-axis). The training score (red) and validation score (green) curves represent performance on the training and validation datasets as a function of the hyperparameter values. **B)** The learning curve illustrates training examples versus accuracy, with the X-axis showing training examples and the Y-axis representing accuracy. A small gap between training and validation scores indicates that the model's complexity is appropriate for the data, avoiding overfitting and ensuring good performance on both sets. **C)** The precision-recall curve plots precision (Y-axis) against recall (X-axis) for various thresholds. Curves that localize closer to the top-right corner indicate better model performance.

considerable effort because of variations in size, distinct color patterns, and geographical distributions. Here, we propose an automated strategy to identify cone snail species using a cohesive machine learning (ML) algorithm framework based on feature-assisted training of the *Conus* shell imaging dataset. Our proposed ML model achieved an accuracy of 95% with an 80:20 train-test data ratio, utilizing 38,080 and 9,520 cone snail shell images, respectively.

To ensure clear feature delineation and consistency, we implemented a preprocessing scheme that included grayscale conversion [56], binary image generation [57], image quality enhancement, and Canny edge detection [35, 58]. Edge detection is a crucial preprocessing step that enhances the visibility of key features for accurate identification [59]. This process refines image comparison and improves feature visibility by employing methods used in image recognition. Here, edge detection supports object segmentation and RF-based recognition, thereby strengthening overall performance [60]. Further preprocessing steps included

background removal [61], quality checks, image transformation [62, 63], and feature extraction using Haralick features [41], deep features [42], color moments, and local binary patterns [39], which collectively enhanced the training dataset's quality.

In this study, we utilized a conventional Local Binary Pattern (LBP) approach combined with additional features, significantly improving the recognition rate compared to LBP variants such as LBP Variance (LBPV) and Center Symmetric LBP (CS-LBP). The integration of these additional features addressed the limitations of conventional LBP and its derivatives. Faudzi and Yahya evaluated four LBP derivatives—conventional LBP, LBP Variance (LBPV), Center Symmetric LBP (CS-LBP), and Completed-LBP (CLBP)—under varying environmental conditions [39]. Their findings suggested that LBPV had a higher recognition rate, while CS-LBP excelled under contrast changes, highlighting that applying conventional LBP with additional features can yield better results.

Next, we employed a genetic algorithm for feature selection. Soltanian-Zadeh et al. utilized a comprehensive methodology to extract features from mammographic images using four distinct methods: shape features, Haralick features, wavelet features, and multi-wavelet features [41]. Our approach mirrored this strategy by leveraging a deep learning model (VGG16) for feature extraction, enabling automated learning of complex shell patterns [64]. Deep learning, particularly through convolutional neural networks like VGG16, facilitates hierarchical feature extraction from image objects [65–67]. For cone snail shell images, which exhibit subtle morphological differences [68], deep learning effectively captures fine details such as shell patterns and color gradients. Jaderberg et al. reported that deep learning techniques significantly enhance recognition accuracy for complex image targets [69]. In this study, we integrated Haralick features with additional features derived from the deep learning model, resulting in a robust and informative feature set that improved accuracy.

The model's efficiency was cross-validated by including data from unrelated species, ensuring that features from other species differed significantly from those of Conus. The species support histogram (to assess the distribution of different species number ranges) demonstrated multiple species with high support values, positively contributing to model efficiency. Additionally, we generated a heatmap to depict the macro average, accuracy, weighted average for recall, precision, and F1 score, revealing the highest weighted precision average of 0.958, indicating improved performance. We observed minimal fluctuations in F1-score values across different species, with a value of 0.76 for *Conus consors*. The Structural Similarity Index Metric (SSIM) results ranged from 0.33 to 0.99, indicating varying levels of structural similarity among individual images. As reported by Zhou et al., SSIM can effectively assess structural similarity and serves as a reliable evaluation tool for image quality assessment [70]. These findings suggest that our proposed model recognized multiple species as positive instances, making it more reliable and scalable than manual feature extraction, particularly for handling large datasets (Fig 10).

Among various classification models, the RF model demonstrated reliable results [44, 71], validating Conus species recognition. The RF approach incorporates random feature selection and serves as an effective tool for high-dimensional complex datasets, ensuring robust classification results [72, 73]. The effectiveness of the RF approach has been proven in various applications, including pattern recognition and species identification [74]. The novelty of our approach lies in integrating deep learning-based feature extraction with a supervised learning RF model. Deep learning captures nuanced details through the image dataset [75], while supervised learning optimizes classification accuracy [76], creating a robust and automated system capable of efficiently handling species recognition tasks.

A thorough analysis of learning and validation curves can inform model selection and parameter tuning. Goriya et al. focused on applying fine-tuned ResNet and DenseNet models

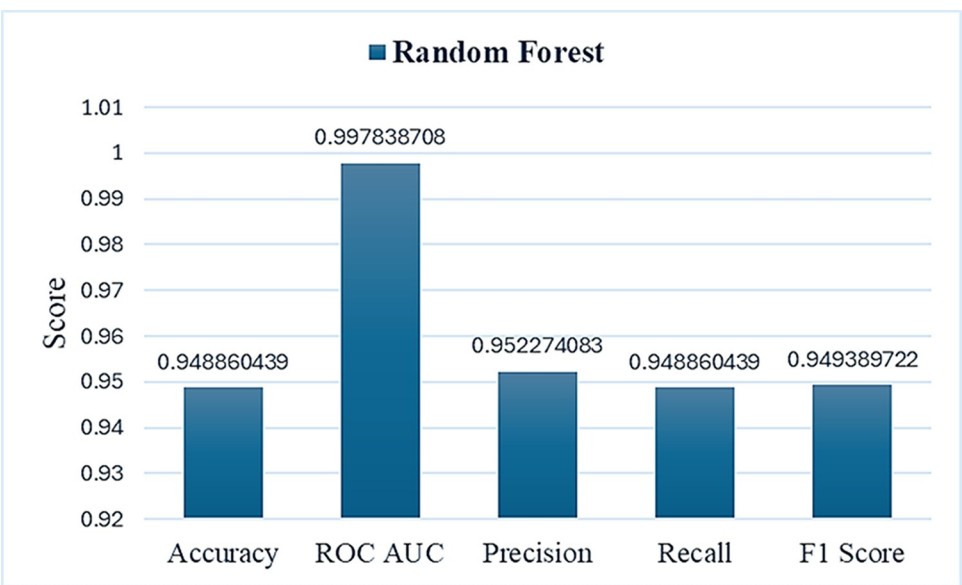

**Fig 10. The overall model metric analysis.** The blue bars represent the average accuracy rates of the model, displaying metrics such as recall, precision, F1 score, and ROC AUC value.

for classifying choroidal neovascularization (CNV) from optical coherence tomography (OCT) images, demonstrating promising results with high accuracy and validation scores [77]. In our study, the DenseNet model achieved a validation accuracy of approximately 0.95, with both training and validation curves exhibiting similar trends. Specifically, our training accuracy reached 99%, while the validation accuracy gradually increased to 95% (Table 2). These values indicate a well-trained model that generalizes effectively to validation data without significant overfitting. This observation suggests that our model, similar to DenseNet, effectively captures the underlying patterns of cone snail shell images through accurate classification. The gradual improvement of the validation score curve is crucial for ensuring model reliability and minimizing the risk of overfitting [77]. The training accuracy of our proposed model resembles the learning curve reported for the RF model by Afuwape et al., which exhibited similar performance metrics [78]. Such similarities in learning curves reinforce the robustness of the RF algorithm in handling classification tasks.

## Conclusion

Overall, machine learning approaches, particularly the Random Forest model, are instrumental in the categorization of cone snail species and in distinguishing them from other marine

**Table 2. The statistical report for RF model evaluation.**

|  | Precision | Recall | F1-Score | Support | TPR | FPR | FNR | TNR |
|---|---|---|---|---|---|---|---|---|
| **mean** | 0.9583 | 0.9560 | 0.9560 | 79.6386 | 0.9560 | 0.0439 | 0.00011 | 0.9998 |
| **std** | 0.0572 | 0.0411 | 0.0405 | 8.5645 | 0.0410 | 0.0411 | 0.00015 | 0.00015 |
| **min** | 0.6346 | 0.8228 | 0.7674 | 37 | 0.8227 | 0 | 0 | 0.9991 |
| **25%** | 0.9426 | 0.9289 | 0.9341 | 74 | 0.9289 | 0.0117 | 0 | 0.9998 |
| **50%** | 0.9762 | 0.9714 | 0.9664 | 80 | 0.9714 | 0.0285 | 0.00007 | 0.9999 |
| **75%** | 1 | 0.9882 | 0.9864 | 84.5 | 0.9882 | 0.0710 | 0.00017 | 1 |
| **max** | 1 | 1 | 1 | 98 | 1 | 0.1772 | 0.00088 | 1 |

invertebrates. The proposed RF model, tested on diverse datasets encompassing both cone snail and other mollusk shells, demonstrates its capability in effective pattern matching and decision-based ranking. This model could also be adapted to detect and classify various other mollusk species, showcasing its versatility and potential for broader applications in marine biology.

## Supporting information

**S1 Fig. Species distribution on the basis of RGB intensities.** Average predicted values were 70.23 for R, 88.12 for average G, and 107.98 for B. **A)** First 59 species (X-axis) with their respective RGB values (Y-axis). **B)** Last 60 species (X-axis) with their respective RGB values (Y-axis).
(TIF)

**S1 Table. Specie prediction results.** Highlighted five rows indicate wrong predictions.
(DOCX)

## Acknowledgments

The authors would like to thank members of the Functional Informatics Lab, National Center for Bioinformatics, QAU, Islamabad for their valuable support.

## Author Contributions

**Conceptualization:** Sajid Rashid.

**Data curation:** Rimsha Bibi.

**Formal analysis:** Noshaba Qasmi, Rimsha Bibi.

**Investigation:** Noshaba Qasmi.

**Methodology:** Noshaba Qasmi.

**Supervision:** Sajid Rashid.

**Visualization:** Rimsha Bibi.

**Writing – original draft:** Noshaba Qasmi.

**Writing – review & editing:** Sajid Rashid.

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
