## [Decision Letter · Decision Letter 0]

11 Sep 2024

PONE-D-24-36557Conus specie recognition through combinatory action of supervised learning and deep learning-based feature extractionPLOS ONE

Dear Dr. Rashid,

Thank you for submitting your manuscript to PLOS ONE. After careful consideration, we feel that it has merit but does not fully meet PLOS ONE’s publication criteria as it currently stands. Therefore, we invite you to submit a revised version of the manuscript that addresses the points raised during the review process.

**the article discuses nice work but I have some comments**:

**the article need some English editing**

**also you need to discuss the literature in a depth and compare it to your work and show the novelty of your work**

**finally, please make sure that the text is following PLOSONE criteria**

**Good luck **

We look forward to receiving your revised manuscript.

Kind regards,

Ramada Rateb Khasawneh

Academic Editor

PLOS ONE

**Journal Requirements:**

**Additional Editor Comments:**

the article discuses nice work but I have some comments:

the article need some English editing

also you need to discuss the literature in a depth and compare it to your work and show the novelty of your work

Reviewers' comments:

Reviewer's Responses to Questions

**Comments to the Author**

1. Is the manuscript technically sound, and do the data support the conclusions?

Reviewer #1: Yes

Reviewer #2: Yes

2. Has the statistical analysis been performed appropriately and rigorously? 

Reviewer #1: Yes

Reviewer #2: Yes

3. Have the authors made all data underlying the findings in their manuscript fully available?

Reviewer #1: Yes

Reviewer #2: Yes

4. Is the manuscript presented in an intelligible fashion and written in standard English?

Reviewer #1: Yes

Reviewer #2: Yes

5. Review Comments to the Author

**Reviewer #1: **The authors proposed "Conus specie recognition through combinatory action of supervised learning and deep learning-based feature extraction". The structure of the article is well structured. But authors will consider following comments.

1.Proofread the entire manuscript.

2.Draw a Graphical abstract of the proposed approach.

3.Compare your approach with previously available existing approaches.

4.Explain the novelty of the proposed approach.

5.Explain why you choose the deep learning based feature extraction.

6. Gold standard dataset is available freely to Conus specie recognition?

7. What type of features played an important role to recognize Conus specie ?

8.compare your approach with previous traditional feature extraction approaches.

**Reviewer #2: **The manuscript can be accepted. I just would like to suggest you include the conclusion heading and try to provide the figures in as higher quality as you can because of the nature of this work. Also, I recommend authors to include some more models rather just being on very simplest model.

6. PLOS authors have the option to publish the peer review history of their article (what does this mean?). If published, this will include your full peer review and any attached files.

Reviewer #1: No

Reviewer #2: No

---

## [Author Response · Author response to Decision Letter 0]

2 Oct 2024

On behalf of all coauthors, I would like to thank you and both reviewers for the valuable feedback to enhance the quality of our manuscript entitled “Conus specie recognition through combinatory action of supervised learning and deep learning-based feature extraction”. 

Reviewer #1

The authors proposed "Conus specie recognition through combinatory action of supervised learning and deep learning-based feature extraction". The article is well structured. But authors will consider the following comments.

1. Proofread the entire manuscript.

We have thoroughly reviewed the manuscript for any grammatical errors and sentence structure anomalies. All sections have been carefully proofread and issues have been resolved. We believe that the manuscript now meets the standards expected for publication in terms of readability. Some changes according to requirements for PLOS ONE:

• Removed the heading numbers, and changed heading 1 to 18, heading 2 to 16, heading 3 to 14.

• Changed in-text citation style from [1][2] to [1,2]. Same for all in-text citations.

• More literature review information related to random forest and imaging was added to the introduction.

• Changed Figure 1 to Fig 1 in both paragraph and caption (did this for all figures).

• Unbold the table reference in paragraph.

• Changed Figure S1 to S1 Fig (same for supplementary table).

• Changed the Fig 6 caption placement right below the paragraph.

• Added the reason for choosing the deep-learning model and the novelty of our approach in the discussion section.

• Add funding as separately according to the requirement.

2. Draw a Graphical abstract of the proposed approach.

We have included a graphical abstract to describe the details of a work plan. 

3. Compare your approach with previously available existing approaches.

Prior to initiate this work, we have thoroughly overviewed previous approaches. Generally, these methods for Conus specie recognition rely on the traditional image processing techniques or features, which have limitations such as sensitivity to image noise and variability in specie appearance. In this study, we utilized hybrid approach that combines deep learning with traditional feature extraction methods followed by supervised learning, feature extraction automation, improved handling of interspecies variability, and noise resilience. These steps resulted in a more reliable and scalable model as compared to conventional methods.

4. Explain the novelty of the proposed approach.

The novelty of our approach lies in the integration of deep learning-based feature extraction with supervised learning. Deep learning captures nuanced details from images, while supervised learning optimizes classification accuracy. This combination allows a more robust and automated system which is capable of efficiently handling specie recognition task adaptable to other biological datasets.

5. Explain why you choose the deep learning based feature extraction.

Deep learning, particularly VGG16, is known to automatically extract hierarchical features through images. For Conus specie shells, which exhibit subtle morphological differences, deep learning-based feature extraction captures fine details such as shell patterns and color gradients. This method is more reliable and scalable than manual feature extraction, especially for large datasets.

6. Gold standard dataset is available freely to Conus specie recognition?

The dataset was mainly extracted from ConoServer database, which is freely available. 

7. What type of features played an important role to recognize Conus species?

Critical features for accurate Conus specie classification included shell patterning, color variations, texture, surface details, and overall shell shape and size. 

8. Compare approach with previous traditional feature extraction approaches.

Feature extraction refers to the process of transforming raw data into numerical features that can be processed while preserving the information in the original data set.

In discussion section, we addressed the rational for using deep learning-based feature extraction method and “Haralick” method. Haralick features are derived from the Gray Level Co-occurrence Matrix. This matrix records how many times two gray-level pixels adjacent to each other appear in an image. Then based on this matrix, Haralick proposes 13 values that can be extracted from the Gray Level Co-occurrence Matrix to quantify texture.

The traditional feature extraction approaches including Principal Component Analysis, Local Binary Patterns (LBP) and Natural Language Processing (NLP)-based techniques are not well defined as image-derived text data is an important source for NLP-based systems. The objects recognized in images (nouns) can be linked to actions (verbs) and attributes (adjectives) described in text, creating a more comprehensive understanding of a scene. The problems are associated with poor alignment of visual and textual data due to differences in data structure and representation. More sophisticated algorithms are required to cope with these issues. Similarly, we cannot use LBP for cone shell detection as it is sensitive to image noise which hinders its ability to accurately capture texture information. The LBP operator compares neighboring pixel intensities, and if there is noise in the image, it can result in incorrect binary values that can affect the resulting LBP histogram.

Reviewer #2

The manuscript can be accepted. I just would like to suggest you include the conclusion heading and try to provide the figures in as higher quality as you can because of the nature of this work. Also, I recommend authors to include some more models rather just being on very simplest model.

We are thankful to worthy reviewer for appreciation. As per your suggestion, we have added a "Conclusion" heading to the manuscript to clearly delineate our findings and their implications. Additionally, we have enhanced the quality of the figures to ensure they meet the standards. 

For image recognition, there are numerous methods which have been proposed recently; however, not all models fit specific tasks due to nature of dataset. There is an issue of overfitting upon dealing with small dataset. The model performs well on the training data but then fails on test data, and lacks performance. The application of advanced neural network structures poses a limitation of their implementation upon architecture variation. Deep learning-based pre-trained models including VGG, ResNet, and Inception are considered as more accurate and efficient in computer vision; however, these models are relatively new and there are certain challenges associated to monitor their benefits. 

We have carefully addressed the comments raised by worthy reviewers. We appreciate the opportunity to revise the manuscript and resubmit it for reconsideration.

Thank you for your valuable time.

---

## [Editor Report · Decision Letter 1]

8 Oct 2024

PONE-D-24-36557R1Conus specie recognition through combinatory action of supervised learning and deep learning-based feature extractionPLOS ONE

Dear Dr. Rashid,

Thank you for submitting your manuscript to PLOS ONE. After careful consideration, we feel that it has merit but does not fully meet PLOS ONE’s publication criteria as it currently stands. Therefore, we invite you to submit a revised version of the manuscript that addresses the points raised during the review process.

We look forward to receiving your revised manuscript.

Kind regards,

Ramada Rateb Khasawneh

Academic Editor

PLOS ONE

Journal Requirements:

Additional Editor Comments:

the article need some English editing

can you discuses the literature more ... and compare your finding with other finding

---

## [Author Response · Author response to Decision Letter 1]

16 Oct 2024

Editor

PLOS ONE 

Dear Editor,

On behalf of all coauthors, I would like to thank you and both reviewers for the valuable feedback to enhance the quality of our manuscript entitled “Recognition of Conus Species Using a Combined Approach of Supervised Learning and Deep Learning-Based Feature Extraction”.

1. Proofread the entire manuscript.

We have thoroughly reviewed the manuscript for any grammatical errors and sentence structure anomalies. All sections have been carefully proofread and issues have been resolved. We believe that the manuscript now meets the standards expected for publication in terms of readability. 

2. Compare current approach with previous approaches.

We have carefully addressed the comments raised by worthy reviewers. We appreciate the opportunity to revise the manuscript and resubmit it for reconsideration.

Thank you for your valuable time.

Sincerely,

Sajid Rashid 

sajid@qau.edu.pk

---

## [Editor Report · Decision Letter 2]

23 Oct 2024

Recognition of Conus Species Using a Combined Approach of Supervised Learning and Deep Learning-Based Feature Extraction

PONE-D-24-36557R2

Dear Dr. Rashid,

We’re pleased to inform you that your manuscript has been judged scientifically suitable for publication and will be formally accepted for publication once it meets all outstanding technical requirements.

Kind regards,

Patrick Goymer

Staff Editor

PLOS ONE

on behalf of

Ramada Rateb Khasawneh

Academic Editor

PLOS ONE
---

## [Editor Report · Acceptance letter]

29 Oct 2024

PONE-D-24-36557R2 

PLOS ONE

Dear Dr. Rashid, 

I'm pleased to inform you that your manuscript has been deemed suitable for publication in PLOS ONE. Congratulations! Your manuscript is now being handed over to our production team.

Kind regards, 

on behalf of

Dr. Ramada Rateb Khasawneh 

Academic Editor

PLOS ONE